# Nanoscale lattice dynamics in hexagonal boron nitride moiré superlattices

S. L. Moore [1✉], C. J. Ciccarino [2], D. Halbertal[1], L. J. McGilly[1], N. R. Finney [3], K. Yao[3], Y. Shao[1], G. Ni[1], A. Sternbach [1], E. J. Telford[1], B. S. Kim[3], S. E. Rossi[1], K. Watanabe [4], T. Taniguchi [4], A. N. Pasupathy [1], C. R. Dean [1], J. Hone [3], P. J. Schuck [3], P. Narang [2] & D. N. Basov [1]

Twisted two-dimensional van der Waals (vdW) heterostructures have unlocked a new means for manipulating the properties of quantum materials. The resulting mesoscopic moiré superlattices are accessible to a wide variety of scanning probes. To date, spatially-resolved techniques have prioritized electronic structure visualization, with lattice response experiments only in their infancy. Here, we therefore investigate lattice dynamics in twisted layers of hexagonal boron nitride (hBN), formed by a minute twist angle between two hBN monolayers assembled on a graphite substrate. Nano-infrared (nano-IR) spectroscopy reveals systematic variations of the in-plane optical phonon frequencies amongst the triangular domains and domain walls in the hBN moiré superlattices. Our first-principles calculations unveil a local and stacking-dependent interaction with the underlying graphite, prompting symmetry-breaking between the otherwise identical neighboring moiré domains of twisted hBN.

[1] Department of Physics, Columbia University, New York, NY, USA. [2] John A. Paulson School of Engineering and Applied Sciences, Harvard University, Cambridge, MA, USA. [3] Department of Mechanical Engineering, Columbia University, New York, NY, USA. [4] National Institute for Materials Science, 1-1 Namiki, Tsukuba, Japan. ✉email: slm2236@columbia.edu

Moiré superlattices have prompted on-demand control of electronic[1–5], optical[5,6], excitonic[5,7,8], plasmonic[9], and magnetic[4,5] properties. A universal attribute of van der Waals (vdW) moiré superlattices is that their emergent physics is demonstrably local with the relevant length scales ranging from atomic to mesoscopic. Accordingly, scanning probe measurements are being widely utilized to visualize the electronic structure and elementary excitations across the moiré landscapes[1,9,10]. Very recently, these efforts have extended to nanoscale studies of lattice dynamics[11,12]. Twist-angle tunability of phonons is an appealing direction in view of their significant role in superconductivity[11], light–matter interaction[13], and thermal transport[14] of vdW heterostructures. Phonons also serve as effective reporters of strain[15] and ferroelectricity[16]. Pursuant to these goals, we report on the lattice dynamics in twisted layers of the prototypical vdW polar insulator hexagonal boron nitride (hBN)[17,18]. Hyperspectral nano-infrared (nano-IR) imaging reveals systematic variations of the in-plane optical phonon frequencies among the triangular domains and domain walls in the moiré superlattices. Our ab initio calculations implicate the underlying graphite substrate in the stark phonon contrast between the moiré domains, hinting at a role of atomic registry in the dielectric coupling between hBN and graphite.

## Results

**Multi-messenger imaging of twisted-hBN moiré patterns**. We used the tear-and-stack method[19] to deterministically produce atomically thin twisted-hBN (t-hBN) moiré superlattices directly on graphite substrates (see Supplementary Note 1 for further description). Nano-imaging data for representative structures are displayed in Fig. 1. We investigated parallel-aligned structures formed by a bilayer on monolayer ("sample 1") and by a monolayer on a monolayer ("sample 2"). It is noteworthy that such parallel stacking is distinct to natural AA′ stacked bilayers. In a past study[12], the lattice dynamics of naturally stacked (AA′) hBN hexagonal moiré patterns demonstrated a small variation of the phonon linewidth within the 500 nm-sized domains. This particular moiré pattern, however, resulted from a happenstance buried interface 15 nm under the surface of a bulk flake, complicating the isolation of the interface lattice dynamics. Here we examine lattice dynamics in the parallel-stacked and atomically thin limits.

The orientations of the Bernal configurations of t-hBN, also known as AB and BA, are defined in Fig. 1a. Unlike AA′ alignment, these configurations host ferroelectric polarization[10,20,21]. Common paraelectric AA stacking occurs when the two hBN layers line up directly above each other. All of these stackings, including the saddle point (SP) configuration (not shown), are represented in equal proportion upon twisting two rigid hBN monolayers. Mechanical relaxation[22] then causes our low-twist-angle samples to form triangular AB and BA domains divided by SP stacking lines with vortices of the domains anchored to AA sites. Our relaxation calculations of parallel-stacked t-hBN in Supplementary Note 5 provide agreement with the triangular features measured here.

We proceed with our study of t-hBN superlattices by visualizing domains and domain walls in t-hBN. Piezo-force microscopy (PFM) imaging in Fig. 1f captures a representative

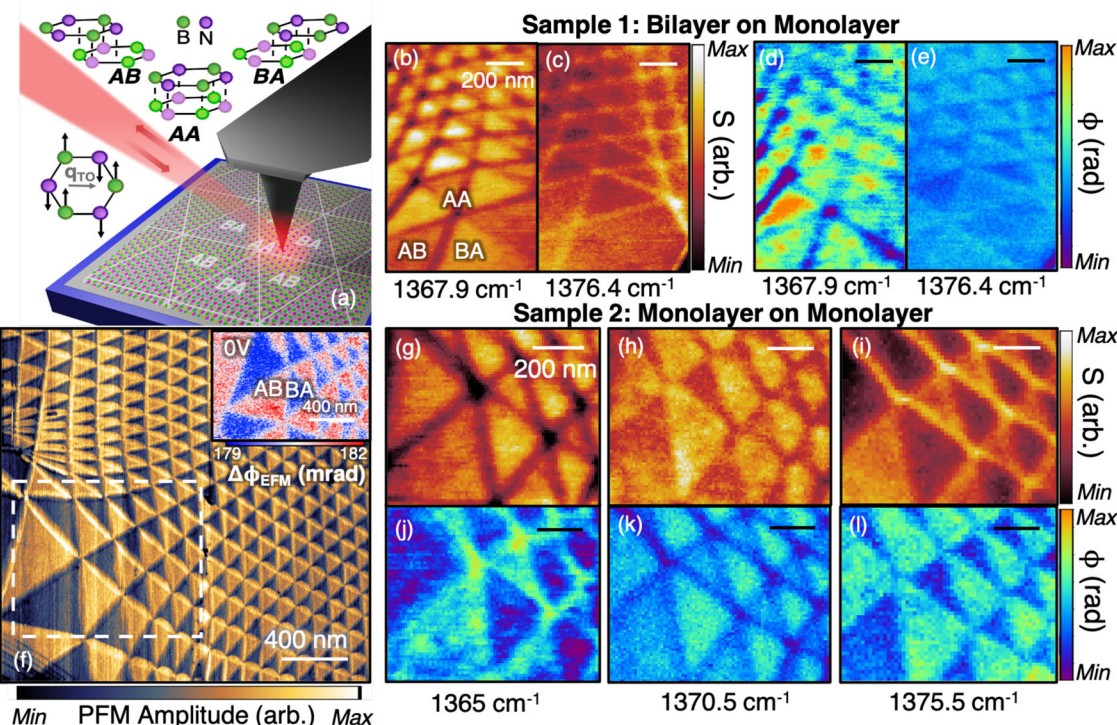

**Fig. 1 Nano-imaging of moiré lattice dynamics in t-hBN. a** Schematic of near-field imaging of t-hBN sample on a graphite/SiO₂/Si substrate. As illustrated by arrows, light scatters off the tip and is sent to a detector. When scanning the tip over the moiré region, we observe a triangular lattice. A schematic of a relaxed superlattice is shown underneath the tip. The possible stacking configurations are drawn above (AA, AB, BA). All experiments are performed with a laser tuned in the vicinity of the transverse optical (TO) phonon frequency. The eigenmode is illustrated on the left, where atoms oscillate orthogonally to the wavevector $q_{TO}$. **b, c** Images of the near-field amplitude taken at two selected frequencies for sample 1 (bilayer–monolayer stacking). **d, e** Images of the near-field phase taken at two selected frequencies for sample 1. **f** Large-area piezo-force microscopy (PFM) image of sample 2 (monolayer–monolayer stacking). The white dashed rectangle shows the region studied with near-field imaging. The inset shows unbiased DC electrical force microscopy (dc-EFM) imaging. **g–i** Images of the near-field amplitude taken at three selected frequencies for sample 2 (monolayer–monolayer stacking). **j–l** Images of the near-field phase taken at three selected frequencies for sample 2.

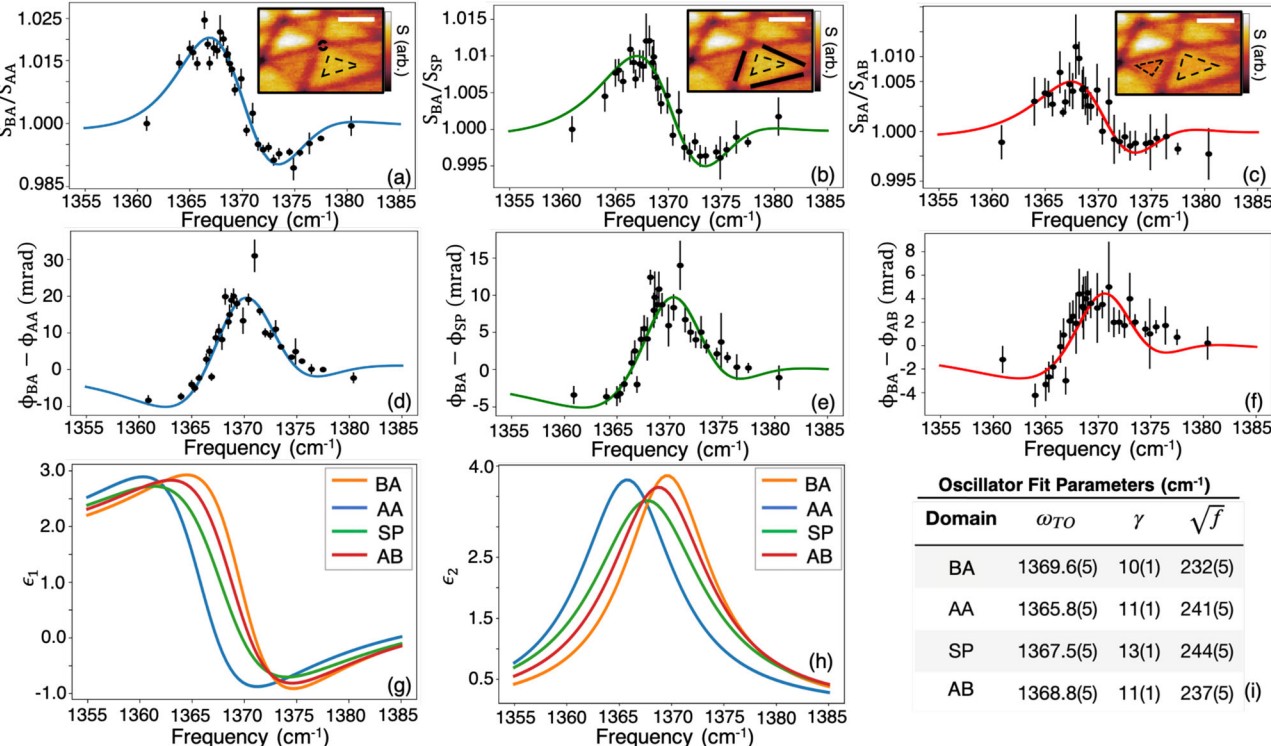

**Fig. 2 Stacking-dependent nano-IR spectroscopy of t-hBN. a–c** Spectra of the scattering amplitude ratios for three independent configurations averaged over AB, BA, and SP regions. The insets are near-field amplitude images of sample 1 at 1367.9 cm$^{-1}$ marking the regions used to construct spectra in the main panels (scale bars are 200 nm). **d–f** Spectra of nano-IR phase differences for three independent configurations averaged over AB, BA, and SP regions. The error bars in **a–f** equal the standard deviation (SD) from averaging amplitude and phase channels across pixels of identical stacking (see "Methods" for more details). All spectra are all simultaneously fit to the Lorentz model (Eq. (1)). The solid lines show the resulting fitting curves described in the text. **g, h** Spectra of the extracted dielectric functions. **i** The fitting parameters, with uncertainties on the last digit in parentheses (e.g., 1369.6(5) = 1369.6 ± 0.5), are reported in the table on the bottom right (all in units of wavenumbers). The uncertainties are taken to be the fitting parameters' SD upon varying the initial guess parameters.

moiré pattern from sample 2. The triangular domains vary significantly in size, a testament to the inhomogeneous twist angle within the field of view of Fig. 1a. PFM contrast is a common reporter of ferroelectric domains[23]. Typically, the contrast reverses between neighboring domains due to a sign-varying ferroelectric polarization. Electrical force microscopy (dc-EFM), also sensitive to the interfacial electrostatic polarization, validates our PFM imaging (inset in Fig. 1f), confirming the ferroelectric nature of the domains[21]. Notably, the curvature of domain walls can be tuned by applying voltages to the tip (Supplementary Fig. 3 of Supplementary Note 3). Positive (negative) tip-sample voltages create an energetic preference for ferroelectric AB (BA) domains, as conjectured by Yasuda et al.[20]. This enables a well-defined assignment to AB and BA domains in our data.

Our phonon-imaging experiments utilize the combined nano-optical and atomic force microscope (AFM)-based scattering-type scanning near-field optical microscopy (s-SNOM), illustrated in Fig. 1a (see "Methods"). Interferometric detection allows us to investigate both the amplitude $S$ and phase $\phi$ of the scattering signal over a broad range of IR frequencies. We focus on the region dominated by the in-plane transverse optical (TO) phonon response between 1360 and 1380 cm$^{-1}$. The s-SNOM imaging provides quantitative topographic information and complex nano-IR contrast in the region of interest with spatial resolution down to 10 nm. As can be seen in Fig. 1b–e, g–l, each element of the reconstructed superlattice—AB/BA domains, AA sites, and SP domain walls—shows a distinct IR contrast. Nano-IR signals (both amplitude and phase) remain relatively uniform throughout either of the two Bernal triangular domains. Meanwhile, we

notice stronger variations among the AA sites and SP domain walls. The overall pattern seen in the nano-IR images follows the shape of the domains uncovered by PFM/EFM, while remaining featureless in AFM topography (Supplementary Note 4). Upon tuning the laser frequency, we detect a strong change, even reversal, of the stacking-dependent near-field amplitude and phase contrast. Such frequency sensitivity suggests systematic spatial variations of the TO resonance between AB/BA domains, SP domain walls, and AA sites (Fig. 1c–e, g–l and additional images in Supplementary Fig. 6 of Supplementary Note 6). The images presented here demonstrate the utility of nano-IR measurements in revealing the trends and real-space patterns in the lattice dynamics of twisted heterostructures.

**Nano-IR spectroscopy of lattice dynamics in twisted hBN**. A salient feature of data in Fig. 1 is a prominent contrast between AB and BA domains. We unambiguously link this contrast to the properties of the TO mode by imaging the domains at a dense set of monochromatic frequencies between ~1360 and 1385 cm$^{-1}$. There, subtle but systematic variations in the frequency-dependent nano-optical contrast are well-captured by the amplitude ratio and phase difference for two stackings (e.g., $S_{AB}/S_{SP}$ and $\phi_{AB}-\phi_{SP}$). This hyperspectral analysis amounts to three sets of independent amplitude and phase data plotted in Fig. 2 for sample 1. Inspecting Fig. 2a–f, we find that the BA/AA and BA/SP contrasts are most prominent. The contrast between AB and BA domains is numerically weaker but highly reproducible. All amplitude spectra show two extrema, whereas phase spectra reveal a single peak centered around 1370 cm$^{-1}$.

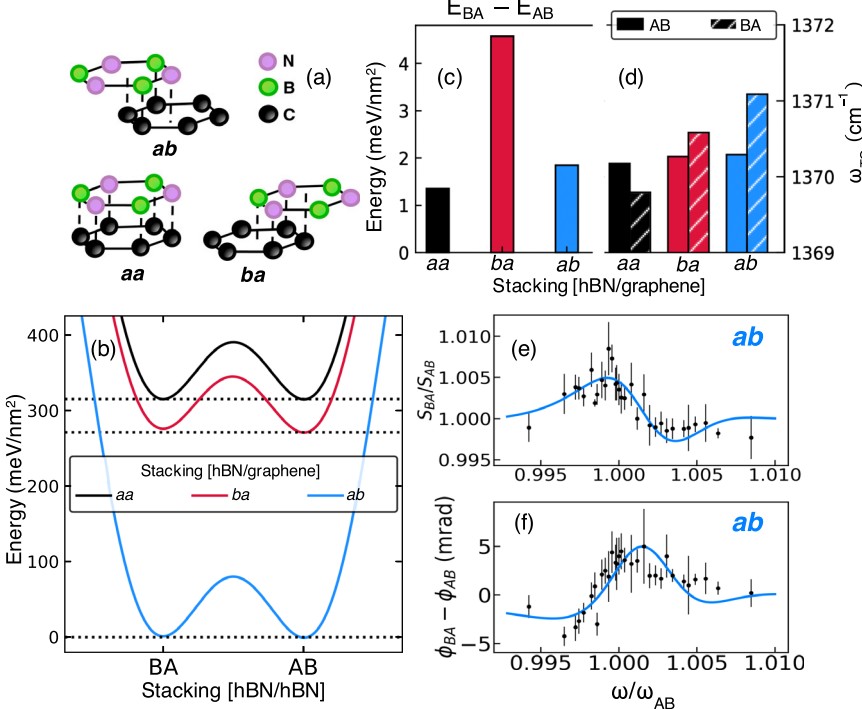

**Fig. 3 Graphite substrate controls the AB/BA contrast in t-hBN.** Uppercase notation (e.g., AB, BA, etc.) is reserved for hBN–hBN stacking configurations, whereas the lower case is utilized to describe various hBN stacking options on graphite (e.g., *aa*, *ba*, etc.). **a** Schematic of hBN/graphite stacking configurations. **b** Stacking energy landscapes for all three commensurate hBN/graphene configurations. Dashed black horizontal lines highlight the subtle energetic preference of AB over BA stackings. **c** Plot of the calculated BA–AB energy difference and **d** TO frequency shift for each commensurate hBN–graphene stacking. **e** Data points reproduce the spectrum of $S_{BA}/S_{AB}$ in Fig. 2c (with the same error bars) plotted using a normalized frequency horizontal axis. Solid line: spectrum of the scattering amplitude calculated with the lightning-rod model. **f** Similar to **e**, data points reproduce the spectrum of $\phi_{BA}$–$\phi_{AB}$ from Fig. 2f. For consistency with Fig. 2, the solid line uses the same oscillator strength and damping (not obtainable in this model) extracted from the Fig. 2i AB fitting parameters.

Spectra in Fig. 2a–f enable quantitative analysis of local lattice dynamics in t-hBN. We describe the optical response by the frequency-dependent in-plane dielectric function $\epsilon(\omega)$. For natural hBN, it is well established that near the TO resonance, $\epsilon(\omega)$ takes the Lorentzian form[24]:

$$\epsilon(\omega) = \epsilon_\infty + \frac{f}{\omega_{TO}^2 - \omega^2 - i\gamma\omega}, \quad (1)$$

where $f$ is the oscillator strength, $\omega_{TO}$ the TO phonon frequency, $\gamma$ the phonon damping rate, and $\epsilon_\infty$ the high-frequency dielectric constant. Equation (1) holds for both bulk and monolayer hBN, but with varying Lorentzian parameters[25,26]. With this dielectric function model on hand, we assume that each sub-diffractional stacking configuration conforms with Eq. (1). We then use the lightning-rod model[27] to fit to our observables: spectra of near-field amplitude $S$ and phase $\phi$. We take $f$, $\omega_{TO}$, and $\gamma$ of each stacking configuration as fitting parameters. The results of the fitting are shown in the solid lines of Fig. 2a–f. Here we also plot the extracted dielectric functions $\epsilon_1 \equiv Re[\epsilon], \epsilon_2 \equiv Im[\epsilon]$ and Fig. 2i tabulates the corresponding Lorentzian parameters. We find a systematic decrease in the TO phonon frequency from the BA to AA configurations by nearly 4 cm$^{-1}$. Similar results for Sample 2 are presented in Supplementary Note 7.

**Origin of the AB/BA TO frequency shift.** To understand the AB/BA frequency shift seen in Fig. 2, we considered the role of strain and also the impact of the graphite substrate (Fig. 3). Our analysis in Supplementary Notes 6 and 7 shows that strain does not extend into the domains themselves and is therefore likely irrelevant to distinctions between $\omega_{TO}^{AB}$ and $\omega_{TO}^{BA}$. We therefore

focus on the role of local atomic registry between hBN and the subjacent graphite. In Fig. 3a, we illustrate three possible hBN/graphite commensurate stackings. For clarity, we label hBN/hBN configurations with the uppercase, e.g., AB/BA, whereas hBN/graphite configurations are denoted in the lower case, e.g., *ab*/*ba*. In Fig. 3a, we illustrate the bottom hBN layer on the top graphite layer for the three possible *ab*, *aa*, and *ba* stackings. The corresponding ab initio stacking energy landscapes are displayed in Fig. 3b. We remark that the AB/BA minima share inequivalent energies in the presence of underlying graphite (Fig. 3c). Calculations reveal that the *ab* alignment is strongly favored over *ba* and *aa* alignments by ~300 meV nm$^{-2}$. The histogram in Fig. 3d shows that graphite's presence also causes a finite frequency shift to the TO mode between AB and BA arrangements. Beyond the qualitative agreement, we find that our calculations for the *ab* stacking arrangement correctly predict the $\omega_{TO}^{BA} - \omega_{TO}^{AB}$ phonon frequency shift. This agreement is illustrated in Fig. 3e, f where we overlay the raw spectra from Fig. 2c, f with the results of nano-IR signal calculations found within the lightning-rod model[27]. Calculations in Supplementary Note 8 reveal a sensitive inverse relationship between the magnitude of the $\omega_{TO}^{BA} - \omega_{TO}^{AB}$ frequency shift and the hBN/graphene separation distance. That frequency shift shows no twist-angle dependence, so long as commensurability is present between the hBN and substrate layers.

**Discussion**

The totality of these observations document that graphite prompts symmetry breaking between moiré domains in hBN. We expect all polar twisted bilayer materials hosting sharp strain-dependent resonances to showcase similar dispersive moiré

patterns. The AB/BA frequency shift should be finite in the presence of any symmetry-breaking substrate, with a magnitude dependent on the sample–substrate interaction. Here, the similar lattice constants between hBN and graphite optimize the effect. Investigations of t-hBN lattice dynamics on other substrates are out of the scope of the present work, but briefly discussed in Supplementary Note 12. The interaction strength between graphene and $p_z$ orbitals of monolayer hBN adds an additional contribution to the harmonic potential landscape and therefore modifies the optical phonon frequency. Similarly, in AB and BA bilayer stacking, the net $z$ dipole of ferroelectric t-hBN interacts with the graphene dependent on the direction of the dipole. This asymmetry is evident from Supplementary Fig. 15, where we calculate the difference in AB/BA dipole moments as a function of graphene separation and local stacking configuration. Thus, although ferroelectricity does not directly soften the phonon, as typical for the out-of-plane modes[28], the ferroelectric dipole's substrate interaction can feasibly translate to modified lattice dynamics between AB and BA stackings.

Empowered with sub-diffractional nano-IR imaging, we resolved nanoscale TO frequency variations in an hBN moiré superlattice. Our first-principles calculations find a new TO frequency shift between otherwise identical domains, dependent on the atomic registry between graphite and atomically thin hBN. Our calculations led us to infer that the directional-dependent interaction of the ferroelectric dipoles with graphite plays a role in the frequency shift of IR phonons associated with the AB and BA domains. These findings are reminiscent of plasmon–phonon coupling between proximal atomic layers of vdW materials[29]. Here, the novelty is that the putative plasmon–phonon hybridization is sensitive to the intricacies of the interlayer atomic arrangements of hBN on graphite. To further test and tune this effect, we anticipate exploration of gated t-hBN devices embedded in plasmonic media.

## Methods

**Sample preparation**. The scotch-tape-exfoliated hBN monolayers used in Samples 1 and 2 are presented in Supplementary Note 1. Sample 1 in Fig. 1a is fabricated via the tear-and-stack procedure with poly-propylene-carbonate, producing a bilayer/monolayer/100 nm graphite sample on top of 90 nm SiO₂/Si substrate. Due to the differing hBN layers here, this sample was produced by first rotating the sample 60° before stacking, achieving a bilayer/monolayer configuration. Sample 2, a monolayer on monolayer, does not require 60° rotation. After stacking, the samples are vacuum annealed up to 500 °C[30]. Empirically, the high temperature seems to remove more tape and polymer residue in the moiré superlattice than lower temperatures and may additionally aid relaxation to *ab* stacking (Supplementary Note 10). The exposed graphite regions after this annealing appear to become rougher, whereas the hBN encapsulated regions remain smooth. We find similar near-field spectra between samples 1 (Fig. 1b–e) and 2 (Fig. 1g–l), suggesting the consistency of the sample preparation technique and relative sensitivity and insensitivity to the interface and total layer number, respectively. This constitutes a more deterministic approach to t-hBN fabrication than previous procedures[12], where a subsurface layer slippage formed in bulk boron nitride after annealing at 1150 °C.

**Piezo-force microscopy**. PFM imaging was performed with a Bruker DimensionFastScan in horizontal deflection mode. The drive amplitude was ~800 mV, which has the effect of controlling the apparent domain wall width and overall contrast. See Supplementary Note 2 for additional images.

**Electrical force microscopy**. We follow the methods described previously[21] to reproduce the ferroelectric contrast in t-hBN. Here, the ferroelectric polarization induces an additional phase shift that can be isolated from topography in a dual-pass lift mode. We find that lift heights <30 nm are sufficient to isolate the EFM phase shift. We use the same microscope system as with PFM. See Supplementary Note 3 for additional images.

**Near-field imaging**. A 20 nm-radius tip of an AFM scans at a 75 kHz tapping frequency over one of our two t-hBN samples. Both structures were assembled on top of ~100 nm-thick graphite and a silicon wafer with ~90 nm-thick SiO₂. An incident, broadly tunable IR laser is chosen to coincide in frequency with the hBN

TO phonon. The laser scatters off of the tip and gets sent to a mercury-cadmium-telluride detector. Upon interferometrically detecting the demodulated signal at harmonics of the tapping frequency (via the standard interferometric pseudo-heterodyne technique[31]), we readily obtain background-free signals with amplitude and phase information. The end result is a collection of nanoscale images in the region of interest: AFM topography, near-field amplitude $S$, and near-field phase $\phi$. The data were acquired with a Neaspec Gmbh microscope with ARROW-EFM PtIr cantilever tips. Our laser is a MirCat quantum cascade laser (QCL) from Daylight Solutions, operating in the range 1360–1380 cm⁻¹. QCL-based nano-spectroscopy offers both superior signal-to-noise and frequency resolution compared with nano-Fourier transform infrared spectroscopy, its broadband-laser-based counterpart. To improve the signal-to-noise ratio, multiple scans were averaged (after correcting for translational drift) in MATLAB at a given frequency. Gwyddion[32], an open-source microscopy package, was then used to compute the statistical properties of each image. Plane leveling and median filtering was consistently used to correct small signal drifts.

**Nano-spectroscopy data analysis**. Using Gwyddion, the near-field responses (amplitude and phase) of identically stacked regions were averaged together at a given frequency. This has the effect of improving the signal-to-noise ratio and rejecting the response from undesirable contaminants. The error bars of the tabulated data, shown in Fig. 2a–f, arise from the (standard deviation) SD. The uncertainties in the lightning-rod model fitting procedure (Fig. 2i) were estimated as the SD of the fitting parameters upon varying the LMFIT python software package's[33] guess parameters. The algorithm's own confidence-interval bounds, meanwhile, were found to overestimate these uncertainties.

**Moiré relaxation simulations**. Modeling of the atomic relaxation of twisted bilayer hBN (presented extensively in the Supplementary Materials) was performed within a continuity model[22] and extended to general real-space geometries[34]. In this model, the total energy of the system is taken as the sum of an elastic energy and a stacking energy term. The total energy was minimized in search for the interlayer real-space displacement field corresponding to the relaxed structure. The stacking configuration at selected points were imposed as boundary conditions. In the periodic case of Supplementary Fig. 5, four such boundary conditions were used to impose a zero external strain condition. In the case of a non-uniform strain map as in Supplementary Fig. 7, such points were chosen to account for the position of AA-stacking configurations from the experimental image. Unless mentioned otherwise, the parameter $\mu$ allocating the displacement field between the two layers was taken as the purely symmetric case of[34] $\mu = 0.5$. The generalized stacking fault energy (GSFE) parameters for the AA-type and AA′-type twisted bilayer hBN heterostructures were taken from Zhou et al.[35], written in the convention of Carr et al.[22]. For AA-type: $c_0 = 8.1678$; $c_1 = 4.8765$; $c_2 = -0.16891$; $c_3 = 0.23089$; $c_4 = c_5 = 0$. All in units of meV/u.c. For AA′-type: $c_0 = 10.7556$; $c_1 = -3.3833$; $c_2 = -0.31262$; $c_3 = 0.11068$; $c_4 = -2.6726$; $c_5 = 0.22816$. All are in units of meV/u.c. Relaxation simulations with domain curvature in Supplementary Fig. 13 of Supplementary Note 9 modify these parameters to accommodate the expected AB/BA energy difference from either *ab* or *ba* BN/graphite stackings. The lattice spacing was taken as 2.51 Å. The bulk modulus $K$ and shear modulus $G$ were taken from Peng et al.[36] and take the values of $K = 95362$; $G = 67106$ in units of meV/u.c.

**Ab initio simulations**. First-principles calculations are performed using density functional theory within the JDFTx code[37]. All calculations are done using ultrasoft pseudopotentials[38] and with the PBEsol exchange-correlation functional[38]. In order to capture the two-dimensional (2D) nature of these systems, we use a Coulomb truncation scheme[39], which successfully removes artificial interlayer interactions in both the electronic optimization and the phonon calculations. This is particularly important, as it correctly predicts no LO–TO splitting in unstrained 2D hBN[40]. The plane-wave basis set used has a kinetic energy cutoff of 30 Hartrees. Phonons are modeled using finite differences and throughout the text we consider the zone center longitudinal optical (LO) and TO mode frequencies, specifically, which are equivalent in the 2D limit. In order to model the vdW interlayer interactions, we use the scheme introduced by Grimme[41]. We use a global scaling factor of 0.15, as this appropriately captures the interlayer spacing of the bilayer hBN, as well as provides good agreement with experiment on optical phonon energies. All calculations use an effective in-plane lattice constant of 2.505 Å.

In the case of calculations related to the substrate-modified stacking energy in Fig. 3, we use a hexagonal unit cell of bilayer hBN stacked on top of graphene. We model the system using a $24 \times 24 \times 1$ Γ-centered $k$-point-mesh and use a $6 \times 6 \times 1$ phonon supercell to determine the harmonic phonon properties. In the case of the frequency shifts with strain, we adopt an orthorhombic unit cell of bilayer hBN for ease of use in applying strain along the in-plane cartesian directions. We use a $18 \times 8 \times 1$ Γ-centered $k$-point mesh and a $6 \times 4 \times 1$ phonon supercell to capture the shifts in the optical modes with strain. We strain the lattice vectors by up to 0.5% in each direction, in order to fit for the A and B constants associated with Supplementary Eq. (2).

Figure 3a shows the three possible orientations that the graphene can adopt with the bottom layer of hBN in the limit that the graphene and hBN lattices are commensurate. In the absence of a substrate, AB/BA stackings are identical, but the

presence of the graphene causes very small differences between the two configurations, both in terms of energy and optical phonon frequency, seen in Fig. 3c.

## Data availability
Raw .gsf files containing the unprocessed near-field images are available from the corresponding author upon reasonable request.

## Code availability
Code used to fit the twisted-hBN contrast is available from the corresponding author upon reasonable request. This code uses the NearFieldOptics python package, downloaded from a github repository from Alexander S. Mcleod.

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

## Acknowledgements
Experimental research at Columbia is solely supported as part of Programmable Quantum Materials, an Energy Frontier Research Center funded by the U.S. Department of Energy (DOE), Office of Science, Basic Energy Sciences (BES), under award DE-SC0019443. C.J.C. and P.N. acknowledge support from the Department of Energy, Photonics at Thermodynamic Limits Energy Frontier Research Center, under Grant number DE-SC0019140. D.H. is supported by a postdoctoral fellowship from the Simons Foundation (579913).

## Author contributions
S.L.M. conducted the sample fabrication, measurements, and data analysis. C.J.C. performed the ab initio simulations. D.H. performed the relaxation simulations. L.J.M. provided expertise on PFM measurements. N.R.F., E.J.T. and B.S.K. provided expertise on sample fabrication. K.Y., Y.S., G.N., A.S. and S.E.R. contributed to the measurements and analysis. K.W. and T.T. grew the hBN crystals. A.N.P., C.R.D., J.H., P.J.S., P.N. and D.N.B. provided expert guidance.

## Competing interests
The authors declare no competing interests.
