## [Peer Review File · Nature Communications]

Nanoscale Lattice Dynamics in Hexagonal Boron Nitride Moiré SuperlatticesREVIEWER COMMENTS

Reviewer #1 (Remarks to the Author):

The work discussed the influence of atomic-scale arrangement on the lattice vibration dynamics in 2D stacked crystals. The twisted stacking of two hBN layers result in different types of Moire lattice, where the electronic and lattice vibration properties will be strongly modified due to the interlayer coupling and formation of superlattice potentials. The authors, as a follow-up study of their previous work (Nat. Commun. 2019, 10, 4360.), manage to demonstrate the influence of local strain on the phonon frequency at the different domains and domain walls. With the help of nano-IR spectroscopy, they show that the presence of graphite substrate, which has a lattice structure similar to that of hBN, can strongly modify the Moire potential of the two hBN layers. Such interaction can lead to inequivalent lattice dynamics of the adjacent AB and BA domains, as manifested from the image contrast of the two domains and systematic shift of the fitted phonon frequency. The experimental findings agree with their ab-initio simulations.

Twisted stacking of vdW 2D crystals have attracted much interest in recent years due to its ability on modifying the electronic, magnetic as well as optical and optoelectronic properties of the materials. The most interesting feature shown in the present study is that presence a substrate with comparable lattice structure with the twisted double 2D layers can strongly affect the bilayer's lattice dynamics. This will in turn modify the physical properties and lead to some possible applications. The ab-intio calculations are clever: they calculated the phonon frequency of hBN bilayer with different stacking individually instead of calculating the whole periodic Moiré superlattice. I think the manuscript is neat and the results deserve publication. Here are a few suggestions and comments.

- 1) As I said, in their previous study the lattice dynamics of twisted bilayers of hBN has been discussed (Nat. Commun. 2019, 10, 4360.). The authors are suggested to give more comments and discussions to differentiate their present study from the previous one.
- 2) If it is possible, it is suggested to provide nano-FTIR spectroscopy on the AB and BA domains, as well as the boundaries to give more quantitative information on the phonon frequency shifts. In addition, AFM topography should be provided.
- 3) For better demonstration, it would be better to mark out the different domains and boundaries in Figure 1b-l.
- 4) In the paragraph below Figure 3, the authors mentioned Figure 3h, but I could not find any related figure.
- 5) According to the calculations, the graphite substrate can heavily influence the lattice dynamics of the stacked hBN layers. Is it possible to perform similar measurements, or calculations, on other types of substrates with lattice structure and constants that are different from the hBN? In this way, the influences from the substrate can be more clearly revealed (it is not obligatory).
- 6) In the end of their discussion, the authors mentioned plasmonic cavity effect from the substrate. However, I could not see any plasmonic effect from their study. Therefore, I suggest the authors either remove or modify this part to make it more relevant to the topic of their study.

Reviewer #2 (Remarks to the Author):

The paper reports the characterization of hBN moiré patterns on graphite, specifically studying the nanoscale infrared response of the hBN stack at various location corresponding to different alignments (AB, BA, AA, SP). Multiple techniques are used (AFM, EFM, PFM, SNOM) and the main result (the asymmetry between the AB and BA configurations) are supported by numerical modelling.

The paper is timely, interesting and correct to the best of my knowledge. It convincingly explains the inequivalence of the AB and BA configurations based on the presence of the substrate that lifts the symmetry, and provides methods that can be applied to other vdW materials and heterostructures. Strikingly, the effects are seen with SNOM and the data is sufficient to fit a Lorentz model for all the alignment cases, revealing different phonon frequencies.

I only have a minor point: the Authors should discuss how they can ensure that the difference in the optical signal in the AB and BA region is truly due to the different optical properties as opposed to being an artifact due to the different mechanical response in the two regions that becomes apparent also in the SNOM data. It would be useful to include AFM images of the sample (without any illumination) to see if any appreciable difference in the signals occurs in the AB and BA regions

Reviewer #3 (Remarks to the Author):

The manuscript used nano-infrared spectroscopy to reveal the lattice responses to the local and stacking-dependent interaction in reconstructed moiré superlattice in the twisted hBN, which is reflected by the frequency variations of the in-plane optical intralayer phonons in triangle domains and domain walls. The authors tried to understand the results by the dielectric function modulated by intralayer phonons. They also investigated the roles of the underlying graphite by the first-principles calculations. The moiré physics in twisted system has attracted great attention while there remains a lack of study of lattice dynamics. Although only two samples are studied and the modulation of moiré superlattice on the phonon dynamics seems not good unveiled, this work would be of potential interest in the field. The authors should carefully clarify several technical issues as follows:

1. The author studied the nano-IR spectroscopy on two samples, bilayer on monolayer and monolayer on a monolayer. I am curious about how the author identify the layer number of hBN flakes. In addition, the hBN flakes with several layer thickness tend to be transparent. Is there any specific method to distinguish the monolayer and bilayer hBN flakes? Can the author present more details about these items? Besides, the authors didn't present the twist angles for these two thBN. Is it possible to check the twist angles from experiment?

2. Upon tuning the infrared laser frequency, the near-field amplitude and phase contrast show different domain distributions, as presented in Fig. 1. Which one corresponds to the real space stacking patterns? Why? What's the reason for such frequency sensitivity? Is this analysis applicable to all the intralayer phonon? Could the analysis extend to other twisted systems? For a new twisted system, how to choose the intralayer phonon for such analysis? These details would be important for one to study the lattice dynamics and moiré physics in twisted systems.

3. In the manuscript, the authors used the nano-IR data to fit the Lorentzian parameters for the sample 1 (bilayer on monolayer) and obtain good fitting results, while the fitting data of sample 2 is shown in the SI. As the stacking of sample 2 is simpler than that in sample 1, it would be easier to see the stacking-dependent lattice dynamics. Would it be better to take the sample 2 as a

prototype? In addition, the authors highlighted that the fitting TO frequency in sample 2 is 1 cm^{-1} larger than that in sample 1, did the author figure out the physics behind this phenomenon? The intralayer phonon in monolayers tends to show Davydov splitting in bilayers, did the authors consider this factor when performing analysis? May the different fitted frequency of TO phonon in AB and BA domain result from the natural AA' stacking in the top bilayer hBN?

4. Are the fitting results from sample 1 applicable to all the samples with different twist angles (e.g., large-twist-angle t-hBN without lattice reconstructions) and/or layer number? Or how can the twist angle and layer number influence the fitting Lorentzian parameters? The authors should give a detailed analysis and discussions on this to enhance the universality of this work.

5. The author demonstrated that the change of the local atomic registry due to the underlying graphite would lead to the frequency variation of TO phonon in AB and BA stacking domains. Did the author try to measure the corresponding nanoscale lattice dynamics on suspended t-hBN to see if they can observe degenerate TO phonon frequency in AB and BA stacking domains? This would provide direct evidence for the effect from underlying graphite substrate on the local atomic registry and thus the frequency variation.

We appreciate both the generous complements and the useful feedback from all reviewers. We are pleased that two reviewers are in favor of publication. We hope that we have provided satisfactory responses to all concerns, adding further discussion in the text where needed.

Reviewer 1

The work discussed the influence of atomic-scale arrangement on the lattice vibration dynamics in 2D stacked crystals. The twisted stacking of two hBN layers result in different types of Moire lattice, where the electronic and lattice vibration properties will be strongly modified due to the interlayer coupling and formation of superlattice potentials. The authors, as a follow-up study of their previous work (Nat. Commun. 2019, 10, 4360.), manage to demonstrate the influence of local strain on the phonon frequency at the different domains and domain walls. With the help of nano-IR spectroscopy, they show that the presence of graphite substrate, which has a lattice structure similar to that of hBN, can strongly modify the Moire potential of the two hBN layers. Such interaction can lead to inequivalent lattice dynamics of the adjacent AB and BA domains, as manifested from the image contrast of the two domains and systematic shift of the fitted phonon frequency. The experimental findings agree with their ab-initio simulations.

Twisted stacking of vdW 2D crystals have attracted much interest in recent years due to its ability on modifying the electronic, magnetic as well as optical and optoelectronic properties of the materials. The most interesting feature shown in the present study is that presence a substrate with comparable lattice structure with the twisted double 2D layers can strongly affect the bilayer's lattice dynamics. This will in turn modify the physical properties and lead to some possible applications. The ab-intio calculations are clever: they calculated the phonon frequency of hBN bilayer with different stacking individually instead of calculating the whole periodic Moiré superlattice. I think the manuscript is neat and the results deserve publication. Here are a few suggestions and comments.

1) As I said, in their previous study the lattice dynamics of twisted bilayers of hBN has been discussed (Nat. Commun. 2019, 10, 4360.). The authors are suggested to give more comments and discussions to differentiate their present study from the previous one.

We thank the reviewer for the suggestion. We have added a discussion on this on pages 1 and 2. Our current work is novel in comparison with the work cited by the reviewer in the following key aspects: First, the previous work studied a buried interface which was the happenstance result of an annealing-driven interface slippage. As a result, the previous work was limited in terms of reproducibility and access to the interface. Here, we present a reproducible approach to realize twisted BN and provide a direct study of the twisted hBN moiré in the atomically thin limit, with much greater details. Secondly, the previous work examined the AA' configuration, which is the opposite alignment from the focus of our current work.

2) If it is possible, it is suggested to provide nano-FTIR spectroscopy on the AB and BA domains, as well as the boundaries to give more quantitative information on the phonon frequency shifts. In addition, AFM topography should be provided.

We thank the reviewer for the suggestion. Unfortunately this is not possible because the required spectral resolution ($< \sim 1 \text{ cm}^{-1}$) is more than what is available on our nanoFTIR interferometer (4 cm^{-1}). We comment in the methods section on page 7 of the revised text that QCL-based nano-

spectroscopy offers both superior signal-to-noise and frequency resolution. Both aspects of the hyperspectral data acquisition were essential for the analysis presented in this work.

3) For better demonstration, it would be better to mark out the different domains and boundaries in Figure 1b-l.

We thank the reviewer for the suggestion. We have added additional labeling in Fig. 1b

4) In the paragraph below Figure 3, the authors mentioned Figure 3h, but I could not find any related figure.

We thank the reviewer for bringing this typo to our attention. "h" has been changed to "e".

5) According to the calculations, the graphite substrate can heavily influence the lattice dynamics of the stacked hBN layers. Is it possible to perform similar measurements, or calculations, on other types of substrates with lattice structure and constants that are different from the hBN? In this way, the influences from the substrate can be more clearly revealed (it is not obligatory).

This is a fantastic suggestion and one that we have been working toward. However, each substrate presents its own difficulties, and therefore warrants a unique study. We present an additional figure in supplementary section 12 showing PFM contrast on hBN and WSe₂ bulk substrates. Due to the sharp resonance on the hBN and the instability to high temperature annealing on the bulk WSe₂, these substrates were not possible to produce a moiré pattern of the necessary cleanliness for SNOM measurements.

6) In the end of their discussion, the authors mentioned plasmonic cavity effect from the substrate. However, I could not see any plasmonic effect from their study. Therefore, I suggest the authors either remove or modify this part to make it more relevant to the topic of their study.

We replaced "plasmonic cavity" with an arguably more common term "plasmon-phonon coupling". As an outlook, we suggest further investigation of such a mechanism by designing gated devices.

Reviewer 2

The paper reports the characterization of hBN moiré patterns on graphite, specifically studying the nanoscale infrared response of the hBN stack at various location corresponding to different alignments (AB, BA, AA, SP). Multiple techniques are used (AFM, EFM, PFM, SNOM) and the main result (the asymmetry between the AB and BA configurations) are supported by numerical modelling.

The paper is timely, interesting and correct to the best of my knowledge. It convincingly explains the inequivalence of the AB and BA configurations based on the presence of the substrate that lifts the symmetry, and provides methods that can be applied to other vdW materials and heterostructures. Strikingly, the effects are seen with SNOM and the data is sufficient to fit a Lorentz model for all the alignment cases, revealing different phonon frequencies.

I only have a minor point: the Authors should discuss how they can ensure that the difference in the optical signal in the AB and BA region is truly due to the different optical properties as opposed to being an artifact due to the different mechanical response in the two regions that becomes apparent

also in the SNOM data. It would be useful to include AFM images of the sample (without any illumination) to see if any appreciable difference in the signals occurs in the AB and BA regions

We thank the reviewer for the comments and suggestion. We have added information in figure S4 of supplementary section 4 showing topography in the moiré regions of samples 1 and 2. The only features are flake boundaries and residues from the sample fabrication process. However, for an AFM instrument of sufficient fidelity, it should be possible to detect the moiré pattern in topography. Typically, topographic height should be largest along domain walls due to relaxation-induced corrugation. In ferroelectrics, polar domains lead to an additional tip-sample interaction, adding to the overall topographic phase contrast. This phase contrast, when isolated with improved signal-to-noise ratio, is the basis for dc-EFM measurements. Our near-field measurements lack the fidelity to observe the moiré directly in topography. Such sensitivity is possible with high-fidelity AFMs and data acquisition modes not compatible with nano-IR spectroscopy/imaging. The best way to verify the pure optical nature of the near-field channel's moiré pattern is with spectroscopy; the highly dispersive imaging of Fig. 1 cannot possibly occur from topographic influences. Rather, we would expect a frequency-independent contrast with no influence from the phonon resonance.

Reviewer 3

The manuscript used nano-infrared spectroscopy to reveal the lattice responses to the local and stacking-dependent interaction in reconstructed moiré superlattice in the twisted hBN, which is reflected by the frequency variations of the in-plane optical intralayer phonons in triangle domains and domain walls. The authors tried to understand the results by the dielectric function modulated by intralayer phonons. They also investigated the roles of the underlying graphite by the first-principles calculations. The moiré physics in twisted system has attracted great attention while there remains a lack of study of lattice dynamics. Although only two samples are studied and the modulation of moiré superlattice on the phonon dynamics seems not good unveiled, this work would be of potential interest in the field. The authors should carefully clarify several technical issues as follows:

1. The author studied the nano-IR spectroscopy on two samples, bilayer on monolayer and monolayer on a monolayer. I am curious about how the author identify the layer number of hBN flakes. In addition, the hBN flakes with several layer thickness tend to be transparent. Is there any specific method to distinguish the monolayer and bilayer hBN flakes? Can the author present more details about these items? Besides, the authors didn't present the twist angles for these two thBN. Is it possible to check the twist angles from experiment?

We thank the reviewer for the suggestion. Supplementary section 1 has been added to describe the monolayer identification process with a high-contrast optical microscope and presents the original microscope images for samples 1 and 2. Ultimately, PFM provides the final confirmation of the layer number, so long as a moiré pattern is observed.

2. Upon tuning the infrared laser frequency, the near-field amplitude and phase contrast show different domain distributions, as presented in Fig. 1. Which one corresponds to the real space stacking patterns? Why? What's the reason for such frequency sensitivity? Is this analysis applicable to all the intralayer phonon? Could the analysis extend to other twisted systems? For a new twisted system, how to choose the intralayer phonon for such analysis? These details would be important for one to study the lattice dynamics and moiré physics in twisted systems.

We thank the reviewer for the question. We are indeed probing the intralayer phonon. The contrast can be understood from the highly frequency-dependent dielectric function presented in Figs. 2j and 2h of the main text and derived from the Lorentz oscillator model (eq. 1). The local dielectric function maps directly to the near-field contrast. The central result of this paper is the finding that the dielectric function near the phonon resonance is stacking dependent. Therefore, as one tunes the frequency, one gets closer to the phonon resonance of some stackings, further from others. As now explained in the main text (page 6), this analysis can be applied to other minimally twisted systems with sharp stacking-dependent resonances of the dielectric function.

3. In the manuscript, the authors used the nano-IR data to fit the Lorentzian parameters for the sample 1 (bilayer on monolayer) and obtain good fitting results, while the fitting data of sample 2 is shown in the SI. As the stacking of sample 2 is simpler than that in sample 1, it would be easier to see the stacking-dependent lattice dynamics. Would it be better to take the sample 2 as a prototype? In addition, the authors highlighted that the fitting TO frequency in sample 2 is 1 cm^{-1} larger than that in sample 1, did the author figure out the physics behind this phenomenon? The intralayer phonon in monolayers tends to show Davydov splitting in bilayers, did the authors consider this factor when performing analysis? May the different fitted frequency of TO phonon in AB and BA domain result from the natural AA' stacking in the top bilayer hBN?

We thank the reviewer for the thoughtful question regarding the 1 cm^{-1} frequency shift between samples 1 and 2. Recall that both data sets quote a 1 cm^{-1} frequency shift between respective AB/BA domains. The AB/BA shift is found from fitting the relative contrast between nearby AB and BA domains, while the 1 cm^{-1} phonon frequency disparity between samples 1 and 2 compares the absolute frequencies between two independent data sets. Thus, sample 2's 1 cm^{-1} higher frequencies are not guaranteed to be physical. There is nonetheless precedence for monolayers showing higher phonon frequencies than AA' bilayers, both in Raman and infrared measurements (Supplementary Refs. 14 and 15). Thus, the AA' bilayer on top of sample 1 may contribute an average of its lower phonon frequency with the phonon frequency from the moiré interface. Supplementary section 7 now includes a discussion to this effect.

We are not aware of reports on Davydov splitting of hBN vibrational modes, only in resonant Raman experiments in TMDs. Likewise, we do not expect it to be a relevant description for our experimental and ab-initio observations. Davydov splitting is defined as an interlayer interaction between identical species in the same unit cell. It occurs between homobilayers, as described extensively in TMDs, but not between sample and substrate layers. Thus, the Davydov splitting of AB and BA stackings should be identical due to their inversion symmetry, since the substrate interaction is disregarded.

4. Are the fitting results from sample 1 applicable to all the samples with different twist angles (e.g., large-twist-angle thBN without lattice reconstructions) and/or layer number? Or how can the twist angle and layer number influence the fitting Lorentzian parameters? The authors should give a detailed analysis and discussions on this to enhance the universality of this work.

We thank the reviewer for the important question. The most important criterion is whether we are in the reconstructed phase (~ 1 degree) or incommensurate phase (>1 degree). Without lattice reconstruction, we would not have the strain contributing to the contrast along the domain walls, nor would we have the necessary substrate alignment to produce the AB/BA frequency contrast. In the reconstructed phase, the domain-wall strain-induced frequency tends to increase as the twist angle decreases (figs. S7 and S9), while the AB/BA frequency remains constant (qualitative contrast in fig. 1 and theoretical predictions in fig. 3). We comment on this on page 6 of the main text and supplementary section 6.

The role of layer number has been discussed already in the previous question: increasing layer numbers washes out the interface moiré response. Beyond what was already discussed, lattice reconstruction also depends on the layer number, as in Fig. S13. In the monolayer/monolayer limit, we have maximal curvature. Meanwhile, the curvature vanishes as the layer number increases to the bulk limit.

5. The author demonstrated that the change of the local atomic registry due to the underlying graphite would lead to the frequency variation of TO phonon in AB and BA stacking domains. Did the author try to measure the corresponding nanoscale lattice dynamics on suspended t-hBN to see if they can observe degenerate TO phonon frequency in AB and BA stacking domains? This would provide direct evidence for the effect from underlying graphite substrate on the local atomic registry and thus the frequency variation.

As also addressed to reviewer 1, we comment that this is a fantastic suggestion and one that we have been working toward. However, each substrate presents its own difficulties, and therefore warrants a unique study. We present an additional figure in supplementary section 12 showing PFM contrast on hBN and WSe₂ bulk substrates. Due to the sharp resonance on the hBN and the instability to high temperature annealing on the bulk WSe₂, these substrates were not possible to produce a moiré pattern of the necessary cleanliness for SNOM measurements.

Summary of Revisions

All revisions are highlighted in red.

- 1) **Revision, requests from the editor:** Data Availability, Code Availability, Author Contributions, and Competing Interests sections have been added to the main text.
- 2) **Revision, Comment 1-1:** The second paragraph of the main text (pgs. 1-2) now includes a discussion about our group's previous work with AA'-type moiré twisted hBN.
- 3) **Revision, Comment 1-2:** A discussion on the merits of QCL hyperspectral imaging is added to the methods section under *Near-field Imaging*
- 4) **Revision, Comment 1-3:** Additional labeling in Fig. 1b is provided
- 5) **Revision, Comment 1-4:** The typo has been corrected, changing "h" to "e"
- 6) **Revision, Comment 1-5:** Supplementary section 12 documents PFM contrast on other substrates. In the second-to-last paragraph of the main text, we discuss the possibility of other substrates for lattice dynamics.
- 7) **Revision, Comment 1-6:** The phrase "plasmonic cavity" has been replaced with "plasmon-phonon coupling" in the conclusion of the main text.
- 8) **Revision, Comment 2-1:** Supplementary section 4 shows topographic height compared with dc-EFM imaging
- 9) **Revision, Comment 3-1:** Supplementary section 1 has been added to show the original flakes before stacking. There, we describe the stacking process.
- 10) **Revision, Comment 3-2:** On page 6, second to last paragraph of main text, we discuss the universality of the AB/BA contrast.
- 11) **Revision, Comment 3-3:** Supplementary section 7 now discusses how an AA' bilayer on top of a monolayer could explain the slight shift in frequency between the samples 1 and 2 data sets.
- 12) **Revision, Comment 3-4:** We address the twist angle dependence of the phonon frequencies in supplementary section 6 and page 6 of the main text.

13) **Revision, Comment 3-5:** Supplementary section 12 documents PFM contrast on other substrates. In the second-to-last paragraph of the main text, we discuss the possibility other substrates for lattice dynamics.

REVIEWER COMMENTS

Reviewer #1 (Remarks to the Author):

I looked carefully throughout the revised manuscript. I found that the authors have addressed my comments and questions well. I therefore recommend acceptance of the revised manuscript.

Reviewer #2 (Remarks to the Author):

The Authors addressed my concerns, and i believe the paper is ready for publication.

Reviewer #3 (Remarks to the Author):

Most comments from three reviewers are addressed point by point, and the corresponding discussions in manuscript has been revised. The authors clearly clarify the difference of sample systems between this work and their previous work, which confirms the novelty and importance of the new work. The unique and advantages of nano- spectroscopy under phonon resonances are further shown and discussed. Also, the doubts on the characterization of layer number of few-layer hBN, along with the universality of the Nano-infrared (nano-IR) spectroscopy and analysis used in this work is fully elucidated. The demonstrations would be adequate if the following minor issues are well addressed.

1) In the manuscript, the authors clarified that the different stacking of different stackings (AB and BA) between the bottom hBN and graphite can cause a finite frequency shift to the TO mode and the calculation confirm this assumption. However, the physical origin for this phenomenon seems not so clear. I suggest the author to give a simple explanation about this from a physical perspective, since the substrate effect is one of the novelties in this manuscript and a more comprehensive understanding is necessary.

REVIEWER COMMENTS

Reviewer #1 (Remarks to the Author):

I looked carefully throughout the revised manuscript. I found that the authors have addressed my comments and questions well. I therefore recommend acceptance of the revised manuscript.

Reviewer #2 (Remarks to the Author):

The Authors addressed my concerns, and i believe the paper is ready for publication.

We appreciate the helpful suggestions from Reviewers 1 and 2.

Reviewer #3 (Remarks to the Author):

Most comments from three reviewers are addressed point by point, and the corresponding discussions in manuscript has been revised. The authors clearly clarify the difference of sample systems between this work and their previous work, which confirms the novelty and importance of the new work. The unique and advantages of nano- spectroscopy under phonon resonances are further shown and discussed. Also, the doubts on the characterization of layer number of few-layer hBN, along with the universality of the Nano-infrared (nano-IR) spectroscopy and analysis used in this work is fully elucidated. The demonstrations would be adequate if the following minor issues are well addressed.

1) In the manuscript, the authors clarified that the different stacking of different stackings (AB and BA) between the bottom hBN and graphite can cause a finite frequency shift to the TO mode and the calculation confirm this assumption. However, the physical origin for this phenomenon seems not so clear. I suggest the author to give a simple explanation about this from a physical perspective, since the substrate effect is one of the novelties in this manuscript and a more comprehensive understanding is necessary.

We appreciate the astute question on the origin of the AB/BA frequency shift. On the simplest level, the frequency shift arises because graphite breaks the inversion symmetry between AB and BA stackings. But the precise value of this frequency shift depends on the lattice matching. The interaction strength between p_z orbitals of monolayer hBN and graphene adds an additional contribution to the harmonic potential landscape and therefore modifies the oscillation frequency associated with the optical mode. Similarly, in AB and BA stacking, the net z dipole of t-hBN interacts with the graphene dependent on the direction of the dipole. This asymmetry is evident from Fig. S15, where we calculate the difference in AB/BA dipole moments as a function of graphene separation and local stacking configuration. This suggests that, while the role of ferroelectricity is not to directly soften the mode, its dipolar-substrate interaction can feasibly translate to modified lattice dynamics between AB and BA stackings.

We add a statement to this regard on page 6, paragraph 2 of the main text. All changes are highlighted in red.

REVIEWERS' COMMENTS

Reviewer #3 (Remarks to the Author):

The authors clarified my concerns, I therefore recommend its publication in the present form.